# Triiodothyronine Acts as a Smart Influencer on Hsp90 via a Triiodothyronine Binding Site

**DOI:** 10.3390/ijms23137150

**Published:** 2022-06-28

**Authors:** Lu Fan, Athanasia Warnecke, Julia Weder, Matthias Preller, Carsten Zeilinger

**Affiliations:** 1BMWZ (Zentrum für Biomolekulare Wirkstoffe), Gottfried-Wilhelm-Leibniz University of Hannover, Schneiderberg 38, 30167 Hannover, Germany; lu.fan@bmwz.uni-hannover.de; 2Department for Otorhinolaryngology—Head and Neck Surgery, Hannover Medical School (MHH), 30625 Hannover, Germany; warnecke.athanasia@mh-hannover.de; 3Institute for Biophysical Chemistry, Hannover Medical School, Carl-Neuberg-Straße 1, 30625 Hannover, Germany; weder.julia@mh-hannover.de (J.W.); matthias.preller@h-brs.de (M.P.); 4Institute for Functional Gene Analytics (IFGA), University of Applied Sciences Bonn-Rhein-Sieg, Von-Liebig-Str. 20, 53359 Rheinbach, Germany

**Keywords:** triiodothyronine, Hsp90, protein microarray, thermophoresis, molecular docking

## Abstract

Microarray-based experiments revealed that thyroid hormone triiodothyronine (T3) enhanced the binding of Cy5-labeled ATP on heat shock protein 90 (Hsp90). By molecular docking experiments with T3 on Hsp90, we identified a T3 binding site (TBS) near the ATP binding site on Hsp90. A synthetic peptide encoding HHHHHHRIKEIVKKHSQFIGYPITLFVEKE derived from the TBS on Hsp90 showed, in MST experiments, the binding of T3 at an EC_50_ of 50 μM. The binding motif can influence the activity of Hsp90 by hindering ATP accessibility or the release of ADP.

## 1. Introduction

The triiodothyronine (T3) levels in blood serum and inside a cell are buffered and stabilized to low nanomolar levels by the hypothalamus–pituitary–thyroid negative-feedback axis and the deiodination of thyronine (T4) [1,2]. The biological action of T3 is mediated by binding thyroid-hormone-binding proteins such as the thyroid hormone receptor (THR) [3]. Cochlear function was reported to be influenced by an aberrant THR with T3-binding deficiency [4]. Elevated or decreased triiodothyronine concentrations are associated with severe disease, because the cell physiological setting of T3 is sensitive to influences that cause clinical symptoms, such as hyper- or hypothyroidism [5,6]. Thyroid hormones have an important function during the fetal and neonatal developmental periods in humans and correlate with possible risks to the human fetus from maternal thyroid disorders, such as hypothyroidism, to the risk of mental retardation in the offspring [7,8]. The actual hormonal conversion is susceptible to genetic perturbations and chemicals; these can lead to different thyroid dysfunctions when the basic building blocks are missing in the diet, as in the case of iodine, or when they are present in the diet, as in the case of thioamides, and there is much evidence that thyroid function can be altered by a variety of chemicals [9,10]. There is a broad understanding of triiodothyronine-binding proteins at the molecular level, so individual therapies focusing on iodine deficiency or hormonal dysregulation exist. Overall, it is still unknown whether other proteins that are indirectly affected also have thyroid-hormone-binding sites. Therefore, we investigated whether triiodothyronine could affect other proteins. Interestingly, it was shown that L-Thyroxine (T4) can enhance thermotolerance in yeast [11]. Hsp90 was selected as a cellular stress marker that forms a broad interactome of the cellular proteome. Because of the key position and susceptibility to active substances, it causes a high vulnerability to cancer cells or other pathogenic cells [12,13,14]. To counteract the intrinsic influence via endogenous factors or hormones, it served as the target to investigate T3 influence using a microarray-based binding assay together with molecular docking and MST experiments.

## 2. Results

A microarray-based binding assay of fluorescently labeled Cy5-ATP was used to analyze the Cy5-ATP binding on Hsp90 without or in the presence of T3 (Figure 1A (left)). This technique has previously been used to identify anti-Hsp90 inhibitors such as Radicicol, which displaced the fluorescently labeled ATP from the binding site [15]. The purified Hsp90a protein was printed in very small spots (diameters in the micrometer range) on an NC-coated glass slide, and the binding of Cy5-ATP was applied. After the removal of the unbound label by washing the slides, the binding of Cy5-ATP could be clearly measured as an intense fluorescence signal. Surprisingly, the presence of T3 during incubation significantly enhanced the Cy5-ATP binding to Hsp90 in a dose-responsive manner (EC_50_ = 70 ± 30 nM; Figure 1A, right). Since it is possible that Hsp90 has a T3 binding site (TBS), molecular docking experiments were performed to predict the preferred binding site of T3 on Hsp90. First, by constructing affinity maps with AutoLigand [16] for the high-resolution crystal structure of the Hsp90 N-terminal domain with bound ATP (pdb: 3t0z) [17], we identified six possible binding pockets in the structure with different volumes, shapes, and physicochemical properties, including the ATP binding pocket (Figure 1B(a)). Subsequent blind docking allowed us to screen for favorable binding of T3 to the entire Hsp90 crystal structure and to confirm that all these pockets were able to accommodate T3. We finally carried out a series of targeted docking experiments with each of the individual binding pockets as search areas using AutoDock Vina [18]. As a result, we found the most favorable binding position (according to the predicted binding affinity) of T3 in a binding pocket formed by helix 5 (H5) and the β8-strand of the antiparallel β-sheet in the Hsp90 structure (Figure 1B). A total of four hydrogen bonds stabilized the binding of T3 in this position—two hydrogen bonds of the carboxy group of T3 with the backbone atoms of Ile214, one hydrogen bond of the amino group in T3 with the backbone of Ser211, and one hydrogen bond between the hydroxyl group of T3 and the backbone of Leu220—as well as hydrophobic interactions with residues Val207, Ile218, Leu220, and the CH2 groups of Lys204 (Figure 1B(b)). Additionally, a halogen bond was found between the iodine substituent of the T3 phenol moiety with Glu200.

Next, to study the affinity for T3, from this sequence motif, a peptide was designed (Tyr1 1_HHHHHHRIKEIVKKHSQFIGYPITLFVEKE), reflecting the identified Hsp90 peptide motif that enabled Cy5 labeling for thermophoresis measurements (MST) by His-tag labeling. Titration with an increasing T3 concentration on the Cy5-labeled Tyr1 peptide gave a dose-responsive binding curve at an EC_50_ of ~146 μM (Figure 1B(c)). This result demonstrated that the identified sequence motif had T3-binding properties; other proteins with T3-binding sites had much higher affinities in the low nanomolar range, but our data indicated that T3 influenced the ATP binding on Hsp90. This may indicate that the compact structure of the full-length pocket structure stabilizes the TBS. A comparison with other T3 binders indicated that this peptide motif may also be present in other known or yet unidentified T3-binding proteins (Figure 1C).

## 3. Discussion

The TBS on Hsp90 is special because it can act as a helix-turn-beta motif. These structures are often found in transcription factors and other highly regulated proteins. This would be an additional way to indirectly modulate Hsp90 by either fixing ATP or preventing its release and would thus give access to many proteins that are folded by Hsp90 [12]. This influence may become relevant to the high T3 concentrations or variety of chemicals in an environment with T3 activity. The cell physiological adjustment to T3 levels is sensitive, and its influence is accompanied by clinical symptoms such as hyper- or hypothyroidism [5,6]. Our results indicate that high T3 concentrations can affect the Hsp90 activity, so decreased Hsp90 activity is expected because ATP remains trapped in the binding pocket. Because of the concentration range and susceptibility, T3 is a smart influencer on Hsp90 activity. Mutations in the ATP-binding site on Hsp90 influence the affinity for geldanamycin; therefore, it is possible that functional conformational changes in HSP90 might be affected by the binding of T3, thereby allosterically affecting the ATP binding site. Consequently, reduced Hsp90 activity is associated with a lower stress-compensation capacity. Unfolded protein aggregates that cannot be removed in time are toxic to cells in the long term [20,21]. On the other hand, the synthesis of T3 itself could also be blocked, because the folding pathways for enzymes that synthesize T3 are disrupted. In addition, the T3 binding site on Hsp90 could also be a target for T3 surrogates or substances, such as those produced by environmental factors that negatively affect protein folding machinery [9].

## 4. Materials and Methods

### 4.1. Materials

Tyr1 peptide encoding HHHHHHRIKEIVKKHSQFIGYPITLFVEKE was purchased from Gen script Biotech, (Rijswijk, NL).

### 4.2. Detection of Microarray-Based Hsp90 Activity

Purified full-length human Hsp90 [15] and Xanthomonas HtpG (XcHtpG) [19] were prepared and transferred into 20 mM Tris-HCl (pH 7.5), 50 mM KCl, 6 mM ß- mercaptoethanol, and 10% (*v*/*v*) glycerol; then, they were spotted on a UniSart^®^ 3D nitro slide (Sartorius Stedim Biotech S.A. 2000125). The proteins (3 mg/mL) were plotted contactless GeSim Nano-PlotterTM (GeSim, Radeberg, Germany) with a nanotip pipette and treated after incubation with a blocking-solution with Cy5-ATP label as previously described [15,19]. T3 (Cayman Chemical; Tallinn, Estonia) was diluted with DMSO and was added into a binding buffer containing 20 mM HEPES-KOH (pH 7.3), 50 mM KCl, 5 mM MgCl_2_, 20 mM Na_2_MoO_4_, 0.01% (*v*/*v*) Tween 20, 2% (*v*/*v*) DMSO, 0.1 mg/mL BSA, and 1 mM DTT at a ratio of 1:50. To exclude the effect of DMSO on the chips, the same ratio of DMSO was added to the binding solution as a positive control and the sub-unit with additional 1 µM radicicol as the negative control.

### 4.3. Molecular Docking

All docking experiments were carried out with the crystal structure of the Hsp90 N-terminal domain with bound ATP (pdb: 3t0z). The structure of T3 was obtained from the pdb data bank. Both the protein and ligand structures were prepared using AutoDockTools [22]. While the ligand was allowed full flexibility during docking, the protein was kept rigid. The identification of potential binding pockets in the Hsp90 structure was performed using AutoLigand [16] on the entire protein structure in a 1.0 Å grid search space with a total volume of 1.69 × 10^5^ Å^3^. The binding pockets were ranked according to the predicted binding energy from initial blind docking with the entire Hsp90 structure. Seven potential binding sites were identified with total volumes between 630 and 2000 Å^3^. The search area for subsequent targeted docking for each individual pocket in the Hsp90 structure was centered in the middle of every individual pocket with a grid spacing of 0.375 Å and a total grid box volume of 2.7 × 10^4^ Å^3^. All docking experiments were performed with Autodock Vina [18] and an exhaustiveness value of 56.

### 4.4. Microscale Thermophoresis Analysis (MST)

Tyr1 and XcHtpG were labeled using a Monolith His-Tag Labeling Kit RED-tris-NTA (NanoTemper Technologies, Munich, Germany). A pre-run was performed in an MST glass capillary and checked at a proper LED power on Monolith NT.115 (NanoTemper Technologies, Munich, Germany), and fluorescence between 400 and 700 counts at final concentrations of 50 nM of Tyr1 and 100 nM of XcHtpG was produced. Different combinations of T3 and constant Cy5-labeled Tyr1 or XcHtpG PBST buffer supplied in the kit were incubated in the dark for 30 min on ice. Then, the samples were transferred into Monolith NT capillaries. The capillaries were inserted into the slots on the sample tray, and the measurements were started with a final constant Cy5-Tyr1 concentration of 50 nM or XcHtpG-Cy5 of 100 nM concentration bound with T3 at different concentrations at a LED power of 10%. Data analyses were performed using NT analysis software. The fluorescence at the 20th second was stable and subject to be fitted to obtain half-maximal effective concentration (EC_50_) values and the graphs. The parameters applied for MST were as follows: power, medium; excitation power, 20%; excitation type, Nano-RED; thermostat setpoint, 22.0 °C.

## Figures and Tables

**Figure 1 ijms-23-07150-f001:**
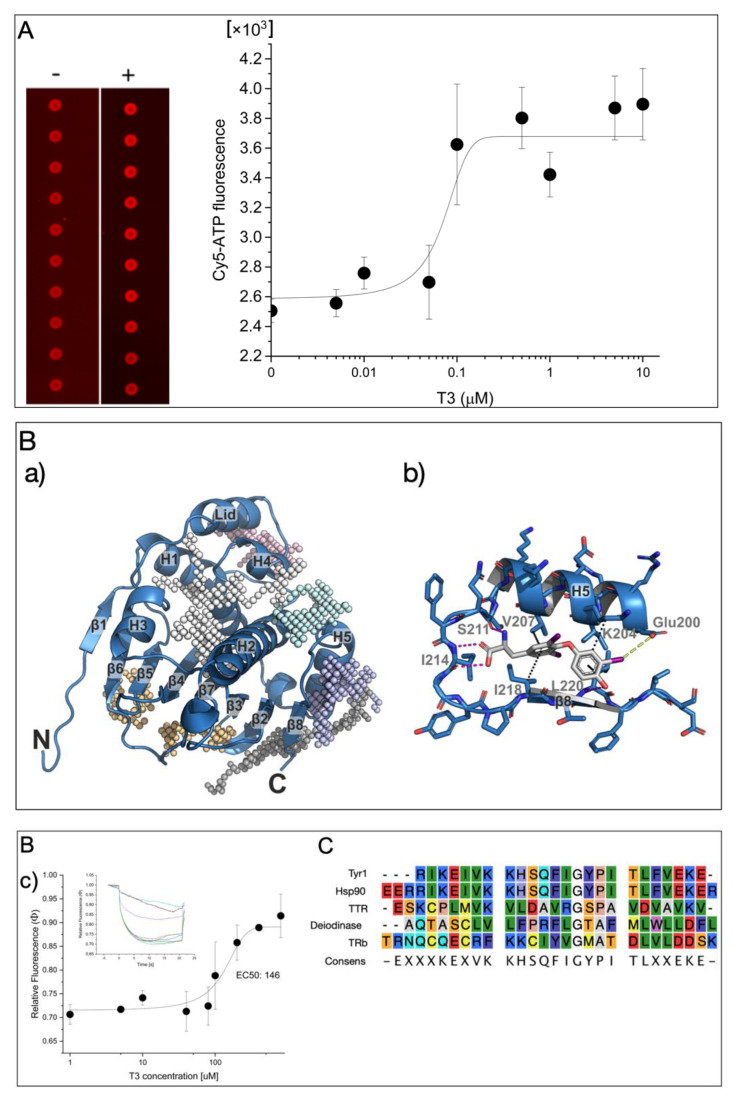
(**A**) Microarray-based analysis of T3 influence on Cy5-ATP binding on Hsp90. Monitoring of bound Cy5-ATP on Hsp90 without and with T3 (left) and corresponding dose-responsive binding activity (right). (**B**) Hsp90 N-terminal domain as binding target of T3: (**a**) overview of the Hsp90 N-terminal domain with identified binding pockets (colored meshes); (**b**) interactions of T3 to the residues of the binding pocket; (**c**) dose-responsive activity of T3 interaction on Tyr1 peptide by microscale thermophoresis analysis (MST). The dose-responsive fittings were performed with function y = A_1_ + (A_2_ − A_1_)/(1 + 10^(LOGx0 − x) × p)^ between the top and bottom asymptotes, with hill slope p and LOGx0 as center at indicated concentration x (inset). MST-traces of Cy5-labeled Tyr1 with increasing concentrations of T3 are displayed in the mode of thermophoresis + T-jump. Different concentrations of T3 are indicated by different colors of traces. Laser-induced temperature changes for F_cold_ were applied from −1 to 0 s, and for F_hot_, from 4 to 5 s. A control experiment with a bacterial HtpG from from *Xanthomonas campestris* [19] with a minor homology in the T3 binding site had no affinity for T3 (Appendix A) (**C**) Amino acid alignment with Tyr1 and Hsp90, Deiodinase, and TTR.

## Data Availability

All raw data formats are available upon publication in IJMS.

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
