# Peer review of "Triiodothyronine Acts as a Smart Influencer on Hsp90 via a Triiodothyronine Binding Site"

_ijms, 2022, doi:10.3390/ijms23137150_

Round 1

Reviewer 1 Report

In the communication entitled “Triiodothyronineis a smart influencer on Hsp90 by a triiodothyronine binding sitethe authors reveal the influence of T3 to Hsp90 and identify the possible binding site of the ligand on the protein using molecular docking. The study is interesting and actual. Some major revisions are needed:

The experiment between Tyr1 and T3 rises the following question: how can one be sure that the peptide conformation is similar and not opposite or quite different to that of the full length protein? I would recommend analogical experiment to be performed with a peptide from the worst predicted binding site, i.e. to perform a negative control analysis for comparison.

It would be essential for the authors to propose an explanation on an atomistic level according to the predicted docking pose of how the binding of T3 to the proposed pocket consisting of H5 and β8-strand, would influence the binding to ATP.

Some minor revisions:

Detailed explanations of the docking protocol, like flexible protein binding site/ligand as well as of the AutoLigand calculations for identification binding sites, like the size of the volume and respectively the number of the series of different volume series need to be provided.

Detailed information of the Hsp90 functions and significance within the cell needs to be included to the introduction section.

Annotations need to be added to all figures.

I would recommend for figure 1B to be more clearly presented (enlarged), as well the ATP binding site to be denoted, maybe with the ligand inserted.

Reference/s for the statements on page 2, line 66 – “This result …… on Hsp90.” need to be inserted.

The sentence started on page 4, line 106: “Different…., 10% (v/v) glycerin and were incubated in darkness for 30 min.” seemes unfinished.

Author Response

Reviewer 1 

In the communication entitled “Triiodothyronineis a smart influencer on Hsp90 by a triiodothyronine binding site” the authors reveal the influence of T3 to Hsp90 and identify the possible binding site of the ligand on the protein using molecular docking. The study is interesting and actual. Some major revisions are needed:

The experiment between Tyr1 and T3 rises the following question: how can one be sure that the peptide conformation is similar and not opposite or quite different to that of the full length protein? I would recommend analogical experiment to be performed with a peptide from the worst predicted binding site, i.e. to perform a negative control analysis for comparison.

Answer: A sequence alignment of human Hsp90a and the bacterial homolog from Xanthomonas campestris showed that the bacterial homolog is quite different to the putative Tyr-binding site of human Hsp90a. Therefore, we concluded that this protein should exhibit no response to T3. The MST control experiment confirmed that T3 was not bound by the Histag Cy5 labelled XcHtpG (S1, MST measurement with purified XcHtpG).

Figure S1: MST-traces of Cy5-labeled XcHtpG with increasing concentrations of T3 are displayed in the mode of thermophoresis + T-jump. Different concentrations of T3 indicate different colors of traces. Laser-induced temperature changes for Fcold were applied from -1 to 0 s, and for Fhot from 4 to 5 s.

It would be essential for the authors to propose an explanation on an atomistic level according to the predicted docking pose of how the binding of T3 to the proposed pocket consisting of H5 and β8-strand, would influence the binding to ATP.

Answer:  For the first time we show that a T3 binding site in Hsp90a exist. On the atomistic level it is too early to fully explain the reason for this effect. It is possible that hydrophobic sites between K209 to Y216 can interact with sites on the H2 helix between E42 to D57, thereby allosterically affecting the ATP binding site. On the other hand, this region is shown to play a role during functional conformational changes of HSP90, which might be affected by binding of T3. However, this is subject of further investigations. Analysis of the ATP-binding site reveal that different sites can influence the ATP binding, whereas exchanges of K48R influence the affinity for geldanamycin. Own mutational experiments reveal that several positions in the pocket are highly sensitive to exchanges and influence the affinity for ATP (manuscript in preparation). Some exchanges can generate changes in the distance and charge and influence thereby the interaction with the phosphates.

Some minor revisions:

Detailed explanations of the docking protocol, like flexible protein binding site/ligand as well as of the AutoLigand calculations for identification binding sites, like the size of the volume and respectively the number of the series of different volume series need to be provided. 

Answer: All docking experiments were carried out with the crystal structure of the Hsp90 N-terminal domain with bound ATP (pdb: 3t0z). The structure of T3 was obtained from the pdb databank. Both, the protein and ligand structures were prepared using AutoDockTools (22). While the ligand was allowed full flexibility during docking, the protein was kept rigid. Identification of potential binding pockets in the Hsp90 structure was done using AutoLigand (16) and the entire protein structure in a 1.0 Å grid search space with a total volume of 1.69 · 105 Å3. The binding pockets were ranked according to the predicted binding energy from initial blind docking with the entire Hsp90 structure. Seven potential binding sites were identified with total volumes between 630 and 2000 Å3. The search area for subsequent targeted docking for each individual pocket in the Hsp90 structure was centered in the middle of every individual pocket with a grid spacing of 0.375 Å and a total grid box volume of 2.7 · 104 Å3. All docking experiments were performed with Autodock Vina (23) and exhaustiveness of 56. 

Reviewer 2 Report

Authors investigated the interactions between Cy5-ATP and Hsp90 without
or in
the presence of T3. The results seems to be very promising, but there are minor corrections to be done:

- resolution of the figures should be improved;

- labels are too small/not readable;

- description of the figures are also missing;

- materials and methods section should be rewritten in more detail;

- it would be helpful to perform short MD refinement and include binding energies of the docking poses.

Author Response

Answer to Reviewer

Reviewer 2

 Detailed information of the Hsp90 functions and significance within the cell needs to be included to the introduction section.

Answer: In the introduction the sentence with references are added. Because of the key position and susceptibility to active substances, it causes a high vulnerability of cancer cells or other pathogenic cells [12-14]. To counteract the intrinsic influence via endogenous factors or hormones it served as the target to investigate the T3 influence using a microarray-based binding assay together with molecular docking and MST experiments.

Annotations need to be added to all figures.

I would recommend for figure 1B to be more clearly presented (enlarged), as well the ATP binding site to be denoted, maybe with the ligand inserted.

Reference/s for the statements on page 2, line 66 – “This result …… on Hsp90.” need to be inserted.

Answer: We agree with this point and revised the sentence: This result demonstrated that the identified sequence motif had T3-binding properties, but other proteins with T3-binding sites had much higher affinities in the low nanomolar range, but our data indicate that the T3 influence the ATP binding on Hsp90.

The sentence started on page 4, line 106: “Different…., 10% (v/v) glycerin and were incubated in darkness for 30 min.” seemes unfinished.

 Answer: The sentence was revised: Different combinations of T3 and Tyr1 were incubated in a buffer containing 20 mM Tris-HCl (pH 7.5), 50 mM KCl, 6 mM ß-mercaptoethanol and 10% (v/v) glycerol for 30 min in the dark.

Authors investigated the interactions between Cy5-ATP and Hsp90 without
or in the presence of T3. The results seems to be very promising, but there are minor corrections to be done:

- resolution of the figures should be improved;

Answer: Pictures with high resolution included

- labels are too small/not readable;

Answer: Labels are emphasized.

- description of the figures are also missing;

Answer Figure legends included.

- materials and methods section should be rewritten in more detail;

Answer: Materials and Methods was formulated in detail.

- it would be helpful to perform short MD refinement and include binding energies of the docking poses.

Answer: This will be done in future and requires more time.

Round 2

Reviewer 1 Report

I accept authors' comments and the communication is ready to be published.